# The recent rapid expansion of multidrug resistant Ural lineage *Mycobacterium tuberculosis* in Moldova

Melanie H. Chitwood [1] ✉, Caroline Colijn[2], Chongguang Yang[3], Valeriu Crudu [4], Nelly Ciobanu[4], Alexandru Codreanu[4], Jaehee Kim [5], Isabel Rancu[1], Kyu Rhee [6], Ted Cohen [1,7] ✉ & Benjamin Sobkowiak [1,7]

The projected trajectory of multidrug resistant tuberculosis (MDR-TB) epidemics depends on the reproductive fitness of circulating strains of MDR *M. tuberculosis (Mtb)*. Previous efforts to characterize the fitness of MDR *Mtb* have found that *Mtb* strains of the Beijing sublineage (Lineage 2.2.1) may be more prone to develop resistance and retain fitness in the presence of resistance-conferring mutations than other lineages. Using *Mtb* genome sequences from all culture-positive cases collected over two years in Moldova, we estimate the fitness of Ural (Lineage 4.2) and Beijing strains, the two lineages in which MDR is concentrated in the country. We estimate that the fitness of MDR Ural strains substantially exceeds that of other susceptible and MDR strains, and we identify several mutations specific to these MDR Ural strains. Our findings suggest that MDR Ural *Mtb* has been transmitting efficiently in Moldova and poses a substantial risk of spreading further in the region.

Multidrug resistant tuberculosis (MDR-TB) is an important driver of TB-related morbidity and mortality and a major threat to TB control in a number of settings. In several countries of the former Soviet Union upwards of 20% of new TB cases have MDR-TB[1], with the vast majority of new cases of MDR-TB resulting from direct transmission rather than through resistance acquired during treatment[2]. Determining the epidemiological and genomic factors that influence the spread of these strains is essential to improve MDR-TB surveillance and control.

The ability for *Mycobacterium tuberculosis (Mtb)* to survive, reproduce, and transmit can vary greatly among drug-resistant strains[3]. Drug resistance in *Mtb* is conferred through chromosomal mutation and is often associated with reductions in bacterial fitness in the absence of the selective pressure of antibiotic treatment. However, the degree and the durability of these fitness costs, as well as the rate of acquisition of resistance-conferring mutations[4], appear to differ by *Mtb* strain and lineage[5–7] and may be mitigated by compensatory mutations[8,9]. For example, strains belonging to the *Mtb* lineage 2 (L2) Beijing sub-lineage have been associated with a higher rate of drug resistance acquisition without a reduction in fitness[10,11], and appear to be a major cause of MDR-TB transmission in many settings. In contrast, the globally dispersed *Mtb* lineage 4 (L4) appears to exhibit more variability in transmissibility between the distinct sub-lineages[12].

We previously reported extensive local transmission of MDR-TB in the Republic of Moldova driven by three large clades of highly drug resistant strains[13]. Two clades of Beijing lineage 2.2.1 strains were found mainly in the semiautonomous region of Transnistria and the third Ural lineage 4.2 clade was present throughout the country. While the Ural lineage has been reported in other countries of the former Soviet Union[14,15], it has not previously been associated with significant local

[1]Department of Epidemiology of Microbial Disease, Yale School of Public Health, 60 College Street, New Haven, CT, USA. [2]Department of Mathematics, Simon Fraser University, 8888 University Drive West, Burnaby, BC, Canada. [3]School of Public Health (Shenzhen), Shenzhen Campus of Sun Yat-sen University, No. 132 Outer Ring East Road, Guangzhou University Town Guangdong, Guangdong, PR China. [4]Phthisiopneumology Institute, Strada Constantin Vârnav 13, Chisinau, Republic of Moldova. [5]Department of Computational Biology, Cornell University, 237 Tower Road, Ithaca, NY, USA. [6]Department of Medicine, Weill Cornell Medicine, 1300 York Ave, New York, NY, USA. [7]These authors contributed equally: Ted Cohen, Benjamin Sobkowiak. ✉e-mail: melanie.chitwood@yale.edu; theodore.cohen@yale.edu

transmission[10]. This apparent widespread transmission of MDR Ural strains in Moldova is a concerning finding.

Here we compare the predicted fitness of MDR and non-MDR Ural and Beijing sub-lineage strains from the Republic of Moldova. Using phylodynamic approaches, we estimate the fitness of clades of MDR and non-MDR strains from both sub-lineages. We identify key genomic differences between these groups to explore putative mechanisms for transmission success of MDR-TB strains in the region. The genomic characterization of MDR-TB strains is critical for the identification of strains with epidemic potential and can aid in the surveillance of such strains throughout the region.

## Results

### Sample data
We use data from a previously reported prospective countrywide study of *Mtb* in the Republic of Moldova[13]. These data include demographic data on all culture-positive TB cases among non-incarcerated adults in the country over the period 1 January 2018 to 31 December 2019 and whole genome sequencing of clinical isolates collected at the time of diagnosis. Of 2770 individuals diagnosed over the study period, 2236 consented to participate in the study and had a *Mtb* isolate available for sequencing. Sequence data was obtained for 2220 patients; 386 individuals were excluded due to evidence of polyclonal infections and 1834 patients were included in the final dataset.

An in silico approach was used to predict lineages and antimicrobial resistance profiles of all strains in the collection[16]. These predictions aligned with phenotypic testing, where available (Supplementary Table 1). There were 804 *Mtb* sequences of the Beijing lineage 2.2.1 (44%), 420 sequences of the Ural lineage 4.2 (23%), and 594 sequences (32%) of other L4 strains. We focused our analyses on Beijing lineage 2.2.1 and Ural lineage 4.2 as these strains were responsible for 97% of the MDR-TB observed in Moldova over the study period. We classify strains as MDR" if the strain is resistant to both Rifampin and Isoniazid and classify all other strains as non-MDR, including strains with monoresistance to first-line treatments (Supplementary Table 2).

### Clade identification
Timed phylogenetic trees were constructed separately for the Beijing lineage 2.2.1 and Ural lineage 4.2 sequences, calibrated by collection dates at the tips (Fig. 1). Next, we looked to identify groups in each tree with distinct demographic or epidemiological histories to conduct phylodynamic analyses and predict clade-level expansions. We used a time threshold to characterize clades that emerged within the last 120 years on the phylogeny, censoring any clades smaller than 20 taxa. The Ural lineage 4.2 taxa were divided into three clades using this time-based approach (Fig. 1A), with two of these clades containing mostly non-MDR strains (Ural B 1% MDR and Ural C 0% MDR), and one clade almost completely comprising MDR strains (Ural A 95.8% MDR). This Ural A clade also includes almost all MDR Ural strains included in the study population (252/256, 98.4%). With the time-based approach, the Beijing lineage 2.2.1 taxa were divided into multiple clades that are more heterogenous than the Ural clades in terms of the proportion of MDR strains (Fig. 1B). We also used an alternative approach to define clades that detects cryptic population structure within phylogenetic trees called treestructure[17]. This model compares the observed ordering of ancestors to the expected ordering in a homogenous population. With this method, we identified population structure in the Ural lineage 4.2 tree (Supplementary Fig. 1A), but not in the Beijing lineage 2.2.1 tree (Supplementary Fig. 1B).

### Markers of clade expansion
To investigate comparative rates of expansion in the Ural and Beijing time-based clades, we estimated the Local Branching Index (LBI)[18] from a maximum likelihood phylogeny produced using the full *M. tuberculosis* dataset, including taxa from both lineages. LBI uses the length of the branches around each internal and terminal node in a phylogenetic tree to estimate local clade expansions, with large LBI values consistent with more rapid branching and clade growth. We found that the average LBI for a terminal node on the tree was 0.018 (range: 0.001, 0.037).

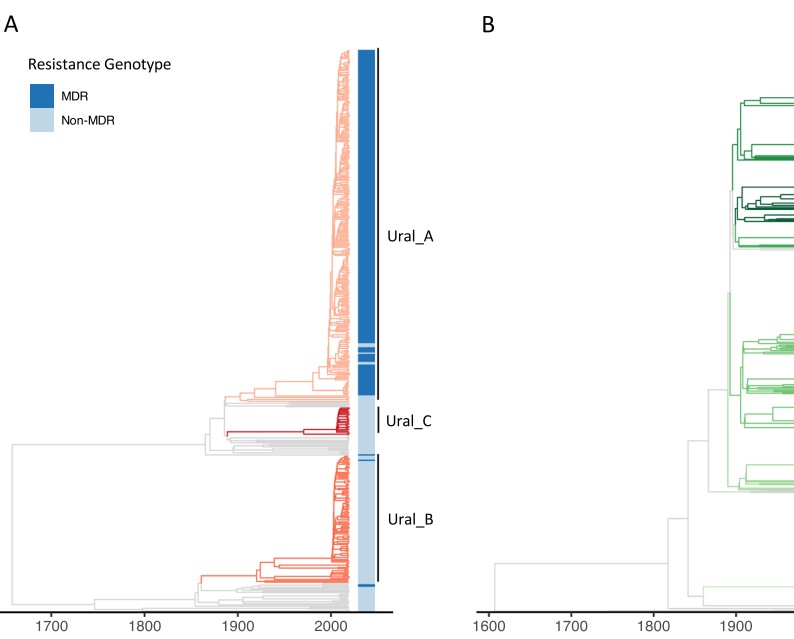

**Fig. 1 | Timed Phylogenetic Trees.** Timed phylogenetic trees of Ural (**A**) and Beijing (**B**) strains used in the study. Internal and terminal branches are colored by the time-based clade designations with corresponding annotated clades names (right). Branches that are not included in designated clades are colored gray. MDR (dark blue) and non-MDR (light blue) phenotypes at the tips are indicated by the colored band. Note that the ancestral phenotype of internal nodes is not inferred in this tree. Source data are provided as a Source Data file.

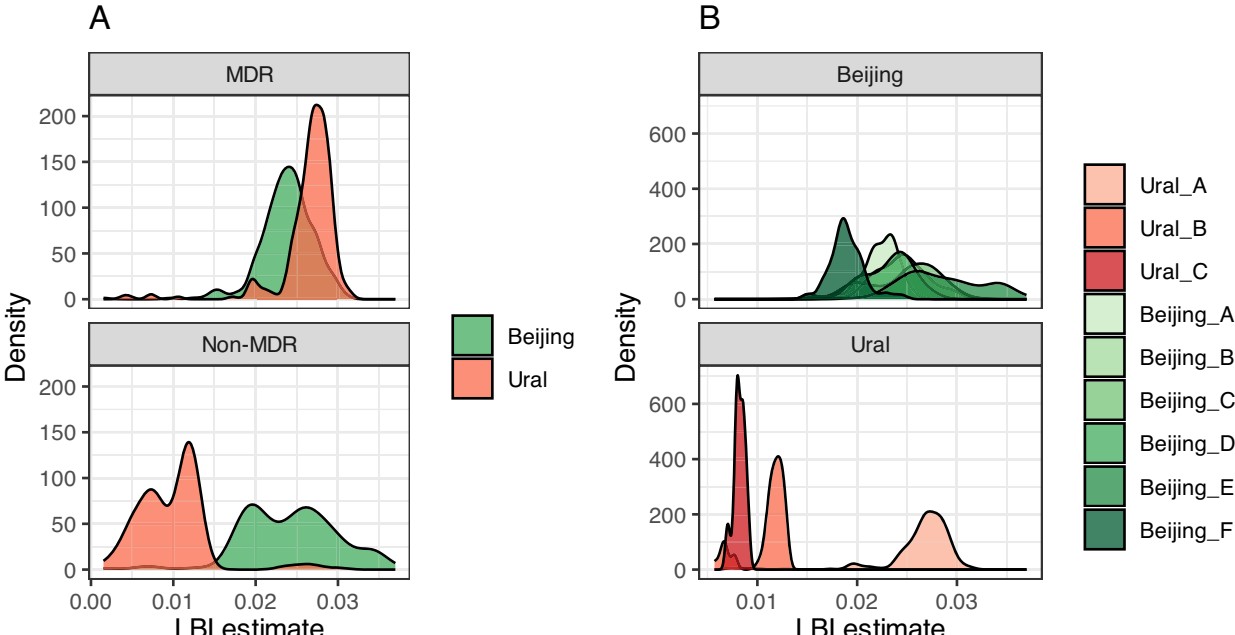

**Fig. 2 | Local branching indices by lineage and phenotype. A** Local branching index (LBI) values for all Beijing (green) and Ural (red) strains used in the study, stratified by taxa phenotype: MDR (top) and non-MDR (bottom). **B** LBI values for strains in each time-based clade, colored by clade (Ural lineage clades are in shades of red and Beijing lineage clades in shades of green) and stratified by clade phenotype: MDR (top, ≥90% of taxa are MDR), mixed (middle ≥10% to <90% MDR), or non-MDR (bottom, <10% of taxa are MDR). Source data are provided as a Source Data file.

We compared the distribution of LBI by phenotype and sublineage and found that the groups differed significantly (ANOVA $p < 0.001$), with post-hoc testing suggesting that the LBI of Ural lineage 4.2 MDR taxa was higher on average than Ural lineage 4.2 non-MDR taxa (Tukey's HSD $p < 0.001$) and Beijing lineage 2.2.1 MDR (Tukey's HSD $p < 0.001$) and non-MDR (Tukey's HSD $p < 0.001$) taxa (Fig. 2A). LBI values for Ural lineage 4.2 MDR taxa were also higher on average than LBIs for taxa belonging to other L4 strains in the collection (Supplementary Fig. 2). Concordantly, there were statistically significant differences in LBI distributions by clade (ANOVA $p < 0.001$). The mean LBI in the majority MDR clade Ural A was 0.026, higher than clades Ural B (mean LBI = 0.011, Tukey's HSD $p < 0.001$) and Ural C (mean LBI = 0.008, Tukey's HSD $p < 0.001$) (Fig. 2B). In contrast, the mean LBI of the MDR clade Beijing A was 0.022, lower than clades Beijing B, Beijing C, and Beijing E; Beijing A had a higher mean LBI than Beijing F (mean LBI = 0.019, Tukey's HSD $p < 0.001$), but the mean LBI did not differ significantly from Beijing D (mean LBI = 0.023, Tukey's HSD 0.79).

Next, we used SkyGrowth[19] to estimate effective population sizes ($N_e$) of each time-based clade separately in the timed phylogenies. This nonparametric method models the growth rate of the effective population size over discrete time intervals, which has a direct relation to the effective reproduction number ($R_e$). We predicted the time dependent $N_e$ between 1990 and the beginning of sampling in 2018 for each clade (Fig. 3). We found that the $N_e$ of the majority MDR clade, Ural A, increased 3.0-fold (95% Credible Interval: 1.30, 5.76) over the period 2010–2018, indicating the recent rapid expansion of this clade. The Ural A increase in $N_e$ was markedly higher than that of the majority non-MDR Ural clades (Ural B 0.26 [−0.45, 2.1] and Ural C −0.29 [−0.84, 2.09]), as well as the majority MDR clade, Beijing A, (1.18 [0.16, 3.12]), over the same approximate time period.

### The effective reproduction number of Ural strains
We estimated the effective reproduction number ($R_e$) of MDR and Non-MDR Ural strains using a multi-type birth death (MTBD) model implemented in the Bayesian phylodynamic inference software BEAST2[20] (Fig. 4). $R_e$ is the expected number of secondary infections

caused by a single infectious individual; a higher $R_e$ suggests a pathogen is spreading more rapidly. Under a birth-death model, new infections are 'born' when a new host is infected, 'die' when the host becomes non-infectious and are sampled with some probability (zero before the beginning of the study period). Infections are stratified into several 'types', and the effective reproduction number of each type is the birth rate divided by the death rate. We found that the predicted $R_e$ of the MDR Ural lineage 4.2 strains was $R_e = 2.5$ (95% Highest Posterior Density: 1.59, 3.80). This was substantially higher than the non-MDR Ural 4.2 strains, with an estimated $R_e = 1.06$ (1.02, 1.16); across model iterations, the $R_e$ of MDR Ural strains was 2.34 (1.55, 3.40) times higher than non-MDR Ural strains.

### Genomic associations with Ural MDR-TB in Moldova
We characterized mutational differences between the MDR Ural isolates and the non-MDR Ural lineage 4.2 and MDR Beijing strains. We identified 70 SNPs and 5 short insertions or deletions (indels) with an allele frequency of ≥90% in the Ural MDR strains that occurred at a frequency of <10% in the non-MDR Ural or MDR Beijing strains (Supplementary Data File 1). Of the 70 SNPs, 37 were non-synonymous mutations in coding regions, 29 were synonymous coding SNPs and four were in intergenic regions. There was evidence of multiple non-synonymous SNPs in genes that have been previously linked to antimicrobial resistance. The *rpoB* S450L mutation, which was found in high proportions in both MDR Ural and Beijing isolates and in low frequencies in non-MDR strains, has been previously well-characterized as conferring rifampin resistance[21]. We also found a high proportion of MDR Ural strains harbored a mutation in *rpsL*, K88R, that has been implicated in streptomycin resistance[22], though with an associated fitness cost[23]. Additionally, a mutation in *ethA*, H281P, was identified in a high proportion of the MDR Ural strains and is almost absent in the rest of the population. This SNP has been observed previously in MDR strains from Moldova[24], and mutations in this gene have been linked to resistance to the second-line TB antimicrobial, ethionamide[25]. Interestingly, we identified a non-synonymous SNP that was present in 98% of our Ural MDR strains in

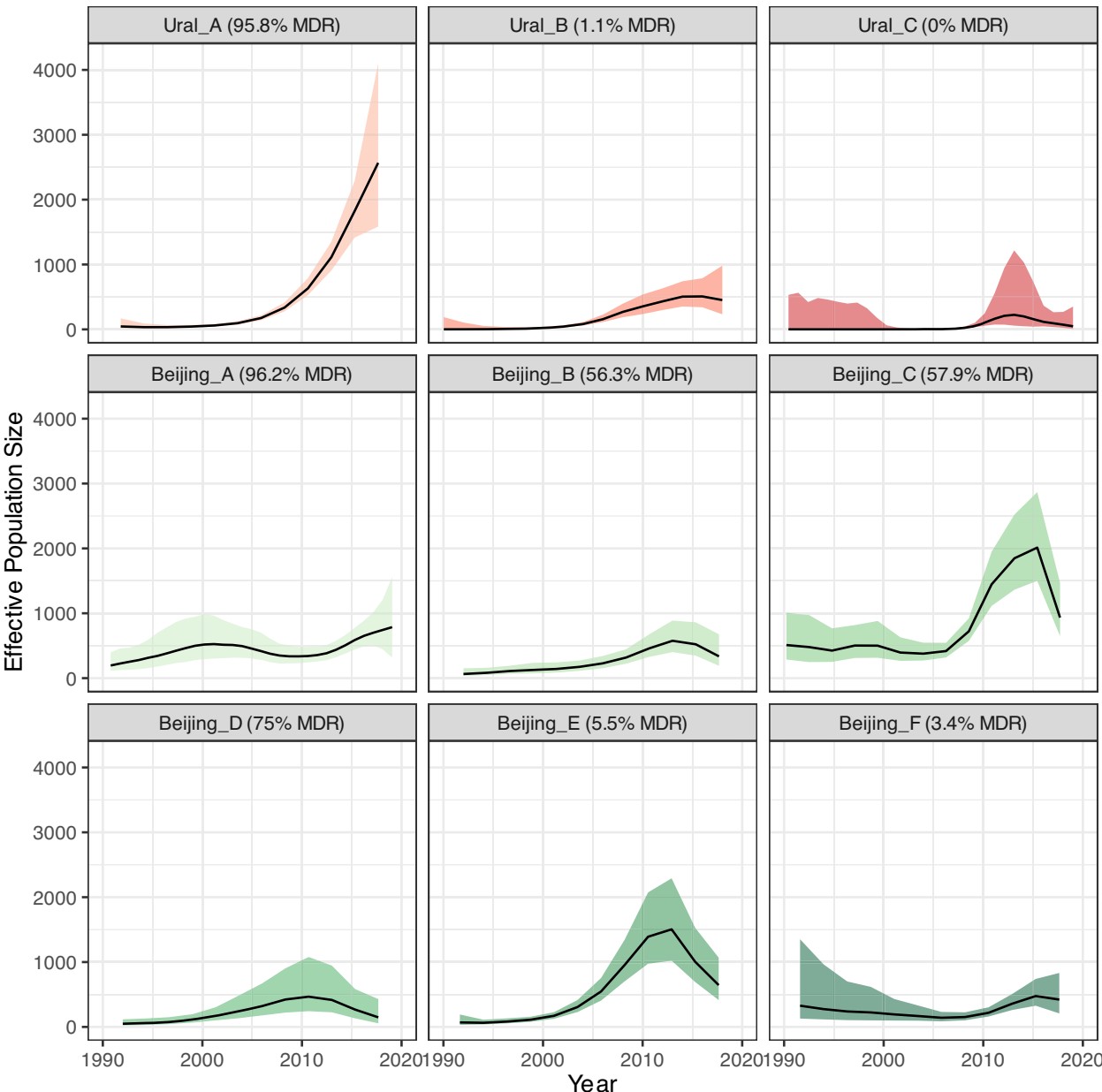

**Fig. 3 | Effective population size estimates by clade.** Effective population size estimates for each time-based clade, inferred using SkyGrowth. Ural lineage clades are in shades of red and Beijing lineage clades in shades of green, with the percentage of MDR strains in each clade shown in the panel header. The line represents the estimated mean, and the shaded area represents the 95% credible interval. Source data are provided as a Source Data file.

*murA*, a key gene in the synthesis of an essential component of the *Mtb* cell wall, peptidoglycan (PG)[26]. A recent study described a previously unknown mechanistic link between PG synthesis and mutations in *rpoB*[27]. The location of the *murA* R110T mutation, near a known active site residue, raises the potential that this may be a compensatory mutation associated with rifampin resistance.

A synonymous SNP in *esxO* at position 2625924 (codon 83), which was fixed in the Beijing lineage, was found to be present in 94% of the MDR Ural strains and only 4% of the Ural non-MDR strains. We also identified a synonymous SNP in *esxP* at codon 10 almost exclusively in the MDR Ural strains. The ESX gene family have been shown to play a role in host-pathogen interaction[28] and we found previous evidence of a mutation in *esxW* in the Beijing strains circulating in Moldova[13]. Synonymous SNPs in the *esxO* and *esxP* have been previously identified as potentially under selection in MDR-TB strains elsewhere[29]. Additional non-synonymous SNPs were identified in genes that may confer a growth or survival advantage, including Rv0355c (*PPE8*) that has been associated with adaptation to host defense mechanisms[30], *mmaA4* previously implicated in drug resistance through inhibition of the mycolic acid biosynthetic pathway[31], and Rv1835c that appears to confer a growth advantage in *Mtb* when disrupted based on transposon mutagenesis[32]. Furthermore, we found five indels at a high frequency in MDR Ural strains that were not present in other isolates, including 2 bp frameshift insertion in the *ndhA* gene, though this has been previously reported to be a non-essential gene for *Mtb* survival[33].

Finally, we performed a genome-wide association study (GWAS) to identify SNPs associated with MDR-TB in both the Ural and Beijing sublineages (Supplementary Table 3). In the Ural strains, we identified only the *rpoB* S450L mutation as significantly associated with MDR strains. We found the same mutation in the Beijing strains, along with the *katG* S315R mutation, which confers isoniazid resistance[34], and the *rpsL* K43R mutation, which confers streptomycin resistance[22].

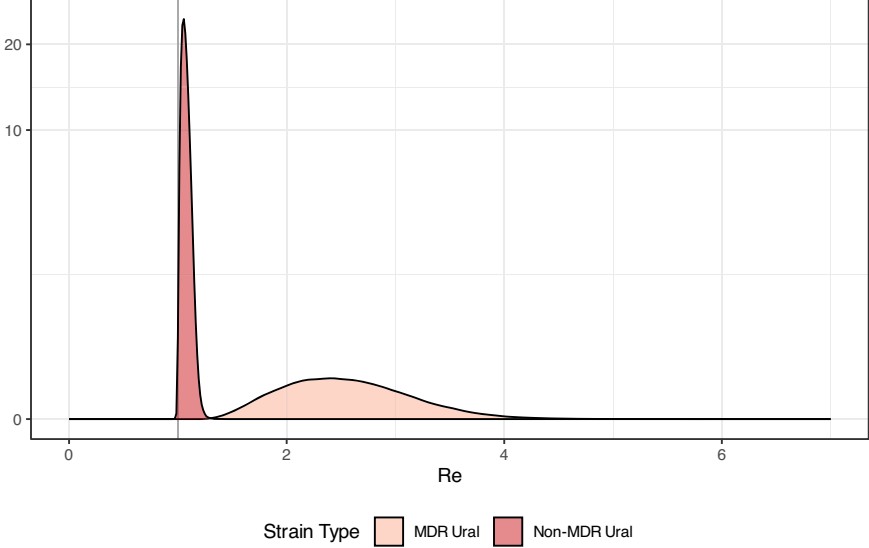

**Fig. 4 | Effective reproduction number estimates from Ural lineage by phenotype.** The effective reproduction number ($R_e$) of MDR (light red) and non-MDR (dark red) Ural strains inferred using a multi-type birth death model in BEAST2. Source data are provided as a Source Data file.

## Evidence of highly transmissible Ural MDR-TB strains outside of Moldova

Ural lineage 4.2 strains have been identified in other countries of the former Soviet Union[10,35] and in light of this, we compared our strains to publicly available *Mtb* Ural lineage sequences from Georgia[10]. We analyzed the Georgian isolates using the same sequence analysis pipeline as the Moldova strains and performed joint variant calling on the Ural strains from Moldova and Georgia. Although the fraction of MDR strains among Ural strains isolated in Georgia was far lower than in Moldova (~6% MDR in Georgia compared to ~61% MDR in Moldova), there was evidence that nine of the 27 MDR Ural isolates from Georgia carried most of the unique mutations present at a high frequency in our MDR Ural strains from Moldova (Supplementary Data File 1). These nine isolates from Georgia were found to be closely related to the Moldovan Ural MDR strains on a phylogeny, suggesting that these highly transmissible strains have beenare also present outside the Republic of Moldova (Supplementary Fig. 3). Of the 70 MDR Ural strain-specific SNPs identified above in Moldovan isolates, 59 (84%) of these SNPs were found in approximately one third of MDR Ural strains from Georgia (9 strains), and four of these strains also harbored a further 7 SNPs (94% of the total SNPs). Two of the known AMR associated SNPs in *rpoB* and *rpsL* were found in higher frequencies in the Georgian MDR Ural strains (78% and 52% respectively) and only two SNPs were not present at all in the MDR Ural isolates from Georgia. All 70 MDR Ural strain-specific SNPs identified in Moldovan samples were found at a low frequency in non-MDR Ural strains from Georgia (<10%).

## Sensitivity Analyses

We conducted several additional analyses to determine whether these results were sensitive to modeling assumptions. To account for the evidence that the average mutation rate in *Mtb* L4 is believed to be slower than in Beijing strains[4], we reconstructed an additional timed phylogenetic tree for the Ural sub-lineage 4.2 with a fixed mutation rate of 0.3 SNPs/genome/year and recalculated effective population sizes. We found that tree could be subdivided into time-based clades that were similar in size and composition to the those identified in the main analysis, and that the majority MDR Ural clade was still predicted to have a faster growth rate than the non-MDR Ural clades and the Beijing clades in the main analysis (Supplementary Fig. 4). We also randomly resampled the Ural sub-lineage 4.2 sequences and re-ran the MTBD model. We found that $R_e$ estimates were consistent across the main analysis and the re-sampled run (Supplementary Fig. 5). Finally, we looked at the demographic composition of the timed clades used in this study to determine whether these factors may contribute to the observed differences in transmission. We found that clades had similar distributions of age, sex, and homelessness (Supplementary Table 4). The Bejing A clade had a higher proportion of formerly incarcerated individuals (24.5%) than the overall average (11.4%), though all other clades did not vary greatly in their fraction of formerly incarcerated individuals (Supplementary Table 4).

## Discussion

We investigated the reproductive fitness of the *Mtb* Ural 4.2 and Beijing 2.2.1 sub-lineage strains responsible for most MDR-TB circulating in the Republic of Moldova. We found that the fitness of MDR Ural strains is high relative to non-MDR strains of the same lineage, and even higher than strains of the Beijing sub-lineage commonly associated with successful transmission. We also found strong evidence that there has been a recent, rapid expansion of a large MDR clade of Ural lineage 4.2 strains in Moldova in the past 10–15 years. Given the high burden of MDR-TB in Eastern Europe, these results suggest that MDR-TB of the Ural 4.2 sub-lineage should be closely monitored in Moldova and the surrounding region to enhance TB control efforts and prevent more widespread transmission. These new findings also suggest the possibility that more intensive contact tracing efforts should be focused on close contacts of individuals with MDR-TB of the Ural 4.2 sub-lineage given our finding of increased reproduction numbers of these strains; implementing such an intervention would require further investment in routine sequencing. We are currently investigating the potential costs and health impacts of routine sequencing in this context[36].

Our findings are consistent with our earlier work in the Republic of Moldova, which highlighted the presence of highly drug resistant Ural lineage 4.2 strains that appeared to be readily transmitting within the population[13,37–39]. However, the $R_e$ estimates for Ural 4.2 strains reported here contrast with a recently published study that found that there was a reduced transmission fitness associated with multi-drug resistance in L4 strains in another former Soviet Union country[10]. This Georgian study, conducted on strains collected two years prior to our work in Moldova, identified only 27 MDR Ural 4.2 strains, and only 14% of the MDR-TB in the country was in L4 strains. Notably, we found that a small number of the Georgian MDR Ural strains shared multiple mutations that were found only in the Moldovan MDR Ural strains in

our study and these Georgian isolates appeared to be closely related to Ural MDR-TB isolates from Moldova. Given the evidence of the recent expansion of MDR Ural strains in Moldova in the past decade, it is possible that MDR Ural sub-lineage strains carrying key mutations that have contributed to widespread transmission in Moldova may now be circulating more widely in Georgia and other countries of the former Soviet Union, though further work is required to determine the fitness effects of these SNPs and their prevalence in the region.

We identified several genes associated with anti-tuberculosis drug resistance in the MDR Ural strains that were not present in high proportions in MDR Beijing strains, including genes conferring resistance to second-line drugs streptomycin and ethionamide. In addition, we identified several mutations that are associated with improved bacterial survival, including in ESX family of genes which are involved in host-pathogen interactions[40]. While these findings provide potential mechanisms by which the relative fitness of the MDR Ural strains is maintained or increased, we only looked for the presence of single variants that were at a high frequency in our MDR strains. A more comprehensive investigation could provide further insights, including the specific fitness effects conferred by individual drug-resistant mutations and putative compensatory mutations, along with an analysis of epistatic interactions between multiple variants present concurrently in MDR strains and the functional modeling of protein changes.

A limitation of this study was that our dataset included *Mtb* isolates collected over a two-year period, making it difficult to observe longer term trends in strain evolution. For this reason, our data are not sufficient to estimate a mutation rate; we used a fixed mutation rate to estimate time-resolved phylogenies. This choice may impact estimates of the effective population size. We reconstructed the timed phylogeny with the Ural strains using a lower fixed mutation rate and found the same pattern of rapid expansion in the majority MDR Ural clade. Furthermore, the reported clade expansion estimates were inferred using clades that were defined by applying a cut-off to the timed phylogeny of 120 years before the study period, removing any tips with a coalescent time to the clade beyond this threshold. As such, these results may be biased by excluding MDR strains with low transmission fitness and by the specific choice of threshold. Even so, the final clades include the majority of Ural and Beijing strains from our dataset and the overall finding, that MDR Ural strains appear to have a high transmission fitness than non-MDR Ural strains, is supported by the LBI estimates that included all strains in the collection. Finally, we excluded 194 (12%) polyclonal infections (resulting from concurrent infection with multiple strains) where at least one of the constituent strains was a Ural lineage 4.2 or a Beijing lineage 2.2 strain. Accurately reconstructing the constituent strains of mixed infections remains a challenge and including these samples as a single consensus sequence may impact the accuracy of the phylogenetic reconstruction and subsequent analyses.

In conclusion, we found that MDR-TB Ural lineage 4.2 strains have significantly higher effective reproductive fitness than non-MDR Ural strains circulating in the Republic of Moldova. There is also evidence of a recent, rapid expansion of a large clade of MDR Ural strains, in contrast to non-MDR Ural and both MDR and non-MDR Beijing clades that do not show the same concerning trajectory. Given that we find that approximately 18% of Ural strains in our study were resistant to fluroquinolones, there is also the risk that the MDR-TB Ural lineage 4.2 strain develops further drug resistance and could evolve to become extensively drug resistant[41,42]. Tracking the evolution and transmission of MDR-TB Ural strains in countries of the former Soviet Union is vital to reduce the burden of MDR-TB in these regions and to implement effective TB control strategies.

## Methods
### Study population and sequence analysis
Ethical approval was obtained for this study from the Ethics Committee of Research of the Phthisiopneumology Institute in Moldova and the Yale University Human Investigation Committee (Number 2000023071). Between January 2018 and December 2019, TB culture-positive individuals in the Republic of Moldova were asked to consent for the use of routinely collected demographic and epidemiological data, and for whole genome sequencing of bacterial isolates. Next generation sequencing was carried out using the Illumina MiSeq platform, with the resulting raw reads mapped to the H37Rv reference strain with BWA 'mem'[43] and single nucleotide polymorphisms (SNPs) called using the GATK software package[44]. We identified polyclonal infections using MixInfect[45] and excluded these samples from further analysis. Lineage typing and in silico drug resistance predictions were carried out with TB-Profiler v2.8.14[16]. Full details on the study enrollment, specimen and individual metadata collection, and whole genome sequencing have been described previously[13]. For the combined analysis of Georgian and Moldovan Ural lineage 4.2 strains, raw sequence data from Ural strains isolated in Georgia10 were obtained from the European Nucleotide Archive (accession numbers PRJEB39561 and PRJEB5058) and mapped to H37Rv as above. Variant calling was re-run using GATK 'GenotypeGVCF' with all Moldovan and Georgian lineage 4.2 isolates jointly to produce a new multi-sequence alignment for the comparative analysis shown in (Supplementary Fig. 3).

### Phylogenetic tree construction and clade identification
A maximum likelihood (ML) phylogeny comprising all isolates was produced from a multiple sequence alignment of concatenated SNPs using RAxML-NG[46] with the 'GTRGAMMA' substitution model. The same approach was used to build a ML phylogeny of the Ural lineage 4.2 strains from Moldova and Georgia, using a lineage 4.1 strain from Moldova as an outgroup. We next constructed time-calibrated phylogenies separately for Beijing lineage 2.2 and Ural lineage 4.2 strains using the R package BactDating[47] and the well-resolved ML tree as input. Timed phylogenies were built using the 'strictgamma' clock model, a root finding algorithm, and a fixed mutation rate 0.5 SNPs/genome/year, which corresponds to previously reported estimates for the *M. tuberculosis* mutation rate[4]. We also conducted a sensitivity analysis by repeating the analysis on the Ural lineage 4.2 strain using a fixed mutation rate of 0.3 SNPs/genome/year to reflect the potentially slower rate reported for Mtb lineage 4. We ran BactDating for $1 \times 10^7$ Markov Chain Monte Carlo (MCMC) iterations, thinning by a factor of 10,000, without a coalescent prior. We assigned taxa to clades with a shared evolutionary history using two different phylogenetic methods. First, we used a time-based method to identify clades with 20 or more tips that emerged within 120 years start of the study period. Second, we assigned taxa to clades based on evidence of an underlying population structure in the phylogeny using the R package treestructure[17].

### Phylodynamic analyses
We estimated the Local Branching Index (LBI) of each tip in the ML phylogeny using the R package TreeImbalance[48]. LBI is a quantitative method to estimate fitness from the shape of phylogenetic tree. LBI estimates are based on the total branch length surrounding each internal and terminal node in a phylogeny, discounting exponentially with increasing distance from the node. Higher values of LBI are associated with rapid expansion and increased fitness. The scale of the exponential discounting, $\tau$, is a function of the relevant neighborhood size around a node. Neher et al. report that $\tau$ should be the average pairwise distance within the tree scaled by a factor of 0.0625[18]. We calculate the LBI for each internal and terminal node based on the ML tree and the scaling factor $\tau$ and summarize results for each terminal node by drug resistant genotype and by clade.

Next, we estimated effective population sizes ($N_e$) using the R package skygrowth[19], a nonparametric autoregressive model to

estimate effective population sizes through time. We fit the model to each time clade separately with a Bayesian MCMC approach and ran the model for 100,000 iterations, allowing estimation of $N_e$ over 50 timepoints prior to the beginning of sampling and a precision factor of 1 standard deviation.

We directly inferred the effective reproduction numbers ($R_e$) for MDR and non-MDR Ural lineage 4.2 strains using a multi-type birth death (MTBD) model[49]. We sampled 200 taxa from the Ural lineage 4.2, maintaining the same proportion of MDR and non-MDR taxa as in the full dataset. The bdmm package[50] in BEAST2[20] version 2.7.4 was used to run the MTBD model using the multisequence alignment of concatenated SNPs and collection dates, accounting for invariant sites. We ran the model for 100 million MCMC iterations or until all parameters had an effective sample size greater than 200, discarding the first 40% of samples as burn-in.

### Genomic analysis

We identified mutational differences between the MDR and non-MDR Ural strains and the MDR Ural and MDR Beijing strains by comparing the allele frequency of all SNPs and short insertions and deletions in these groups. We examined genomic differences between the MDR Ural strains from Moldova and known antimicrobial resistance mutations were annotated using in-house lists based on associated variants described previously[16,51] (Supplementary Data File 2). Finally, we conducted a GWAS to find SNPs associated with MDR in both the Ural and Beijing strains using treeWAS[52].

### Reporting summary

Further information on research design is available in the Nature Portfolio Reporting Summary linked to this article.

## Data availability

The genomic data from the Republic of Moldova used in this study are available in GenBank under accession code PRJNA736718[13]. The genomic data from the Republic of Georgia used in this study are available in European Nucleotide Archive (ENA) at EBI under the accession codes PRJEB39561 and PRJEB5058[10]. The H37Rv reference genome used in this study is available from GeneBank under accession code PRJNA57777. Source data are provided with this paper (https://doi.org/10.5281/zenodo.10783518)[53]. Source data are provided with this paper.

## Code availability

No previously unreported custom computer code or algorithms were used in this analysis. Code to fit the multi-type birth death model are available (https://doi.org/10.5281/zenodo.10779413)[53].

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

## Acknowledgements
This publication was made possible by grant number R01AI147854 (T.C.) from the National Institute of Allergy and Infectious Diseases (NIAID), grant number P01AI159402 (T.C., K.R., J.K., B.S.) from the NIAID, and grant number UL1 TR001863 (MHC) from the National Center for Advancing Translational Science (NCATS), a component of the National Institutes of Health (NIH). Its contents are solely the responsibility of the authors and do not necessarily represent the official views of NIH.

## Author contributions
Conception and design (M.H.C., C.C,. T.C., B.S.), data acquisition (V.C., N.C., A.C., T.C.), analysis (M.H.C., B.S., I.R.), interpretation of results (M.H.C., C.C., C.Y., J.K., K.R., B.S., T.C.), drafting (M.H.C., B.S., T.C.). All authors substantially revised the manuscript and approved the submitted version.

## Competing interests
The authors declare no competing interests.
