## [Peer Review File · Nature Communications]

REVIEWER COMMENTS

Reviewer #1 (Remarks to the Author):

Excellent phylogenetic and phylodynamic analysis of the MDR-TB epidemic in Moldova. The results highlight that, in addition to the well-known Beijing strains, other strains can result in clonal expansion of drug resistant TB in different parts of the world. The work presented serves as an example of how using state-of-the-art phylodynamic methodologies can increase our understanding of the molecular epidemiology of drug resistant TB. The work also identifies some mutations that could serve as a biological mechanism for the observed rapid expansion of the MDR-TB Ural strain. The hypotheses generated can be used to guide future research to better understand Mtb transmission. The limitations of the work are well stated.

Main comments

-386 (17%) individuals were excluded due to evidence of polyclonal infections. Can the authors describe these polyclonal infections? Were the polyclonal strains due to within host micro-evolution or were these mixed infections? Was this due to heteroresistance, mixed infection with two drug resistant strains, mixed infection with two susceptible strains, ... Were polyclonal infection frequent in both MDR-TB and non-MDR-TB strains? Excluding 17% of the strains could bias the results. Did you perform a sensitivity analysis including the majority strain of these 368 patients?

-It is strange that *rpoB* S450L mutations were found in non-MDR strains. Were there patients with rifampicin mono strains? If yes, would it not have been better to classify strains as rifampicin resistant vs rifampicin susceptible rather than MDR-TB vs non-MDR-TB? Especially given that you sometimes refer to the non-MDR as 'susceptible strains'.

-In the introduction, the authors claim that "Determining the epidemiological and genomic factors that influence the spread of these strains is essential to interrupt the transmission of MDR-TB" and "The genomic characterization of MDR-TB strains is critical for the identification of strains with epidemic potential and can aid in the surveillance of such strains throughout the region". Can the authors make a statement on the impact of their findings and suggest how the TB control program of Moldova (and region) could improve their surveillance and control of drug resistant TB based on the findings of the study?

Minor comments:

Introduction:

-Rapid diagnosis and appropriate treatment are the most essential interventions to interrupt transmission, not determining the epidemiological and genomic factors that influence the spread of these strains. Do agree with what is stated later in the introduction, that "the genomic characterization of MDR-TB strains is critical for the identification of strains with epidemic potential and can aid in the surveillance of such strains".

Results

-Is it correct that in Fig 1, the first occurrence of MDR TB dates back to about the 1920s, well before TB treatment was introduced?

-“Comparing LBIs of terminal nodes, we found that the LBIs of Ural lineage 4.2 MDR taxa were higher on average than Ural lineage 4.2 non-MDR taxa and Beijing lineage 2.2.1 taxa (ANOVA $p < 0.001$) (Figure 2A)”. Why did you use an ANOVA test to compare means. It does not look like all LBI distributions follow a normal distribution. I also do not understand the statistical comparisons you did. You compared Ural lineage 4.2 MDR taxa with Ural lineage 4.2 non-MDR taxa and found significant differences in LBI. It is not clear which comparison you refer to in the latter part of the sentence “.... and Beijing lineage 2.2.1 taxa”.

-The legend is clear for Fig 1A as this figure aims to visually demonstrate the differences in LBIT between MDR and non-MDR strains, stratified by Ural vs Beijing. Fig 2B is much harder to read as you refer in the text to different taxa (Ural_A, Ural_B and Ural_C) but it is not clear in the figure which is which. Same issue holds for Fig 1B for Beijing.

-It is interesting that several Beijing clades (C and E) decrease in effective population size, even though Beijing C is predominately MDR-TB. Do you have an explanation for this observation? This is not touched upon in the discussion section.

-While the terminology used in “ ‘die’ when the host recovers” are standard in epidemiological mathematical modelling jargon, suggest you replace ‘the host recovers’ by ‘the host becomes non-infectious’ for a more general public.

-Please write HPD in full when first used

-Figure 4 is not referred to in the text

-‘an essential component of the Mtb cell wall’. Please correct spelling error

-‘we performed a genome-wide association study (GWAS) to identify SNPs associated with multiple drug resistance’ Please change ‘multiple drug resistance’ to MDR-TB (if this is what you want to refer to) as these terms have different meanings.

-You compared the Moldova strains to publicly available Mtb sequences from Georgia. Can you create a phylogenetic tree with both the Moldova and Georgia Ural strains? This would allow readers to visually assess how similar/different these are, probably more intuitive and informative than the current description given in the text.

Reviewer #2 (Remarks to the Author):

The manuscript reads well and provides interesting information, particularly concerning MDR-TB Ural lineage strains.

This work is original and it provides results in settings that have been poorly studied until now.

I do, however, have a few comments to make:

Please check for typo errors:

e.g. "These predictions aligned with phenotypic testing where available (Supplementary Table 1)." in the beginning of the Results section.

Regarding the Results section, in the subsection named "Markers of clade expansion", would it be possible to add the values of LBIs

when mentioning this index (e.g. adding the value of the mean LBI)?

Response to reviewer comments

The authors would like to thank both reviewers for their thoughtful feedback. Below, please find our point-by-point response.

Reviewer #1:

Main comments

-386 (17%) individuals were excluded due to evidence of polyclonal infections. Can the authors describe these polyclonal infections? Were the polyclonal strains due to within host micro-evolution or were these mixed infections? Was this due to heteroresistance, mixed infection with two drug resistant strains, mixed infection with two susceptible strains, ... Were polyclonal infection frequent in both MDR-TB and non-MDR-TB strains? Excluding 17% of the strains could bias the results. Did you perform a sensitivity analysis including the majority strain of these 368 patients?

These 386 individuals carried mixed infections, as discussed in *Yang et al.* While we agree that including these strains would be ideal, the phylodynamic methods used in this study require a single genomic sequence per strain. Additionally, our group and others are currently working on methods to better infer the constituent strains of mixed infection; at present there are no well-supported approaches for accurately reconstructing these sequences. We feel that carrying out a sensitivity analysis including these strains (with a single sequence per mixed sample) would bias the results further by impacting the accuracy of the phylogenetic reconstruction.

Below is a breakdown of the mixed strains (note: the percentage of MDR Ural and Beijing mixed strains is approximately the same percentage as in the pure strains of each lineage):

Mixed strains	Total number	MDR	RIF mono-resistant	INH-mono-resistant
Ural	94	60 (64%)	0	11
Beijing	67	28 (42%)	1	6
Ural/Beijing	16	16 (100%)	0	0
Ural/other	5	5 (100%)	0	0
Beijing/other	12	2 (17%)	0	2

We have included new text in the Discussion section to explain the rationale behind our decision to exclude these strains:

“Finally, we excluded 194 (12%) polyclonal infections (resulting from concurrent infection with multiple strains) where at least one of the constitution strains was a Ural lineage 4.2 or a Beijing lineage 2.2 strain. Accurately reconstructing the constituent strains of mixed infections remains a challenge and including these samples as a single

consensus sequence may impact the accuracy of the phylogenetic reconstruction and subsequent analyses.”

-It is strange that rpoB S450L mutations were found in non-MDR strains. Were there patients with rifampicin mono strains? If yes, would it not have been better to classify strains as rifampicin resistant vs rifampicin susceptible rather than MDR-TB vs non-MDR-TB? Especially given that you sometimes refer to the non-MDR as ‘susceptible strains’

Thank you for raising this concern. Yes, there are a small number of Rifampin mono-resistant cases (4 Beijing and 1 Ural). To improve clarity, we have changed ‘susceptible strains’ to ‘non-MDR’ where it appears in the text. We have also included new text to clarify how we classify strains, and a supplemental table that shows the rates of resistance to first line drugs, stratified by sublineage and MDR classification:

“We classify strains as “MDR” if the strain is resistant to both Rifampin and Isoniazid and classify all other strains as “non-MDR”, including strains with monoresistance to first-line treatments (Supplementary Table 2).”

Supplementary Table 2:

	Beijing MDR (N=394)	Beijing Non-MDR (N=410)	Ural MDR (N=256)	Ural Non-MDR (N=164)	Overall (N=1224)
Rifampin					
No	0 (0%)	406 (99.0%)	1 (0.4%)*	163 (99.4%)	570 (46.6%)
Yes	394 (100%)	4 (1.0%)	255 (99.6%)	1 (0.6%)	654 (53.4%)
Isoniazid					
No	0 (0%)	345 (84.1%)	0 (0%)	127 (77.4%)	472 (38.6%)
Yes	394 (100%)	65 (15.9%)	256 (100%)	37 (22.6%)	752 (61.4%)
Pyrazinamide					
No	105 (26.6%)	408 (99.5%)	188 (73.4%)	163 (99.4%)	864 (70.6%)
Yes	289 (73.4%)	2 (0.5%)	68 (26.6%)	1 (0.6%)	360 (29.4%)
Ethambutol					
No	40 (10.2%)	385 (93.9%)	17 (6.6%)	160 (97.6%)	602 (49.2%)
Yes	354 (89.8%)	25 (6.1%)	239 (93.4%)	4 (2.4%)	622 (50.8%)
Streptomycin					
No	2 (0.5%)	327 (79.8%)	1 (0.4%)	153 (93.3%)	483 (39.5%)
Yes	392 (99.5%)	83 (20.2%)	255 (99.6%)	11 (6.7%)	741 (60.5%)

*low-coverage, likely hetero-resistance at this site

-In the introduction, the authors claim that “Determining the epidemiological and genomic factors that influence the spread of these strains is essential to interrupt the transmission of MDR-TB” and “The genomic characterization of MDR-TB strains is critical for the identification of strains with epidemic potential and can aid in the surveillance of such strains throughout the region”. Can the authors make a statement on the impact of their findings and

suggest how the TB control program of Moldova (and region) could improve their surveillance and control of drug resistant TB based on the findings of the study?

Thank you for this comment. We have added new text to the Discussion:

“These new findings also suggest the possibility that more intensive contact tracing efforts should be focused on close contacts of individuals with MDR-TB of the Ural 4.2 sub-lineage given our finding of increased reproduction numbers of these strains; implementing such an intervention would require further investment in routine sequencing. We are currently investigating the potential costs and health impacts of routine sequencing in this context.”

Minor comments

Introduction:

-Rapid diagnosis and appropriate treatment are the most essential interventions to interrupt transmission, not determining the epidemiological and genomic factors that influence the spread of these strains. Do agree with what is stated later in the introduction, that “the genomic characterization of MDR-TB strains is critical for the identification of strains with epidemic potential and can aid in the surveillance of such strains”.

Thank you for this comment. We agree that the sentence should be reframed, and we have edited it accordingly: “Determining the epidemiological and genomic factors that influence the spread of these strains is essential to improve MDR-TB surveillance and control.”

Results

-Is it correct that in Fig 1, the first occurrence of MDR TB dates back to about the 1920s, well before TB treatment was introduced?

While the time to most recent ancestor (TMRCA) of Ural clade A, which is majority MDR-TB, appears to be around 1900 according to the tree in Figure 1, this does not mean that these ancestral strains were MDR. Instead, drug resistance would have evolved one or more times preceding the sampled tips. A formal ancestral state reconstruction analysis of the resistance-conferring mutations could estimate when these likely evolved, though this is not in the scope of this paper.

We agree that this could be confusing, so we have changed the figure legend to clarify that the MDR and non-MDR colored band illustrates the phenotype at the tips only, not the internal branches and nodes:

“Timed phylogenetic trees of Ural (A) and Beijing (B) strains used in the study. Internal and terminal branches are colored by the time-based clade designations with corresponding annotated clades names (right). MDR (dark blue) and non-MDR (light

blue) phenotypes at the tips are indicated by the colored band. Note that the ancestral phenotype of internal nodes is not inferred in this tree.”

-“Comparing LBIs of terminal nodes, we found that the LBIs of Ural lineage 4.2 MDR taxa were higher on average than Ural lineage 4.2 non-MDR taxa and Beijing lineage 2.2.1 taxa (ANOVA $p < 0.001$) (Figure 2A)”. Why did you use an ANOVA test to compare means. It does not look like all LBI distributions follow a normal distribution. I also do not understand the statistical comparisons you did. You compared Ural lineage 4.2 MDR taxa with Ural lineage 4.2 non-MDR taxa and found significant differences in LBI. It is not clear which comparison you refer to in the latter part of the sentence “.... and Beijing lineage 2.2.1 taxa”.

Thank you for this comment. The ANOVA test itself does not require normally distributed input data, only normally distributed residuals (see histogram of residuals, below). We have edited this section significantly, clarifying which comparisons were made and that the multiple pairwise comparisons were made using the post-hoc Tukey HSD test.

“We compared the distribution of LBI by phenotype and sublineage and found that the groups differed significantly (ANOVA $p < 0.001$), with post-hoc testing suggesting that the LBI of Ural lineage 4.2 MDR taxa was higher on average than Ural lineage 4.2 non-MDR taxa (Tukey’s HSD $p < 0.001$) and Beijing lineage 2.2.1 MDR (Tukey’s HSD $p < 0.001$) and non-MDR (Tukey’s HSD $p < 0.001$) taxa (Figure 2A). LBI values for Ural lineage 4.2 MDR taxa were also higher on average than LBIs for taxa belonging to other L4 strains in the collection (Supplementary figure 2). Concordantly, there were statistically significant differences in LBI distributions by clade (ANOVA $p < 0.001$). The mean LBI in the majority MDR clade Ural A was 0.026, higher than clades Ural B (mean LBI = 0.011, Tukey’s HSD $p < 0.001$) and Ural C (mean LBI = 0.008, Tukey’s HSD $p < 0.001$) (Figure 2B). In contrast, the mean LBI of the MDR clade Beijing A was 0.022, lower than clades Beijing B, Beijing C, and Beijing E; Beijing A had a higher mean LBI than Beijing F (mean LBI = 0.019, Tukey’s HSD $p < 0.001$), but the mean LBI did not differ significantly from Beijing D (mean LBI = 0.023, Tukey’s HSD 0.79).”

-The legend is clear for Fig 1A as this figure aims to visually demonstrate the differences in LBIT between MDR and non-MDR strains, stratified by Ural vs Beijing. Fig 2B is much harder to read as you refer in the text to different taxa (Ural_A, Ural_B and Ural_C) but it is not clear in the figure which is which. Same issue holds for Fig 1B for Beijing.

We agree that the stratification in Figure 2B could be clearer, especially given that specific clades are referenced in the text that are not labeled in the figure. We have updated Figure 2 such that each facet is a phenotype and that each sublineage or clade is colored differently:

Figure 2. (A) Local branching index (LBI) values for all Beijing (green) and Ural (red) strains used in the study, faceted by taxa phenotype: MDR (top) and non-MDR (bottom). (B) LBI values for strains in each time-based clade, faceted by clade phenotype: MDR (top, $\geq 90\%$ of taxa are MDR), mixed (middle $\geq 10\%$ to $< 90\%$ MDR), or non-MDR (bottom, $< 10\%$ of taxa are MDR).

-It is interesting that several Beijing clades (C and E) decrease in effective population size, even though Beijing C is predominately MDR-TB. Do you have an explanation for this observation? This is not touched upon in the discussion section.

Thank you for this question. Yes, there are some clades where we estimate a decrease in effective population size in more recent years. We interpret this to mean that there is more limited spread of these strains. The decrease in effective population size for Beijing C specifically likely indicates that transmission is not currently driving expansion of this MDR clade.

-While the terminology used in “ ‘die’ when the host recovers” are standard in epidemiological mathematical modelling jargon, suggest you replace ‘the host recovers’ by ‘the host becomes non-infectious’ for a more general public.

Thank you, we have made the suggested edit.

-Please write HPD in full when first used

Thank you, we have made the suggested edit.

-Figure 4 is not referred to in the text

Thank you noticing this omission; we have included a reference to Figure 4.

-‘an essential component of the Mtb cell wall’. Please correct spelling error

Thank you, we have fixed this typo.

-‘we performed a genome-wide association study (GWAS) to identify SNPs associated with multiple drug resistance’ Please change ‘multiple drug resistance’ to MDR-TB (if this is what you want to refer to) as these terms have different meanings.

Thank you, we have made the suggested edit.

-You compared the Moldova strains to publicly available Mtb sequences from Georgia. Can you create a phylogenetic tree with both the Moldova and Georgia Ural strains? This would allow readers to visually assess how similar/different these are, probably more intuitive and informative than the current description given in the text.

Thank you for this suggestion. We have included a phylogeny of Ural strains from Moldova and Georgia as a supplemental figure. Interestingly, we found that the nine Georgian strains carrying similar mutations did not appear to be on the same genetic background as the Moldovan strains. We have edited text in the Results and Discussion sections to reflect these new findings.

In Results:

“Interestingly, these nine Georgian strains did not appear on the same genetic background as the Moldovan strains and were paraphyletic with other MDR and non-MDR Georgian isolates, suggesting there was independent acquisition of these SNPs (Supplementary figure 3).”

In Discussion:

“Notably, we found that a small number of the Georgian MDR Ural strains shared multiple mutations that were found only in the Moldovan MDR Ural strains in our study. However, these nine MDR Georgian isolates did not cluster with the Moldovan MDR strains in a phylogeny, suggesting there has been convergent evolution at these loci. Given the evidence of the recent expansion of MDR Ural strains in Moldova in the past decade, it is possible that MDR Ural sub-lineage strains carrying key mutations that have contributed to widespread transmission in Moldova may now be circulating more widely in Georgia and other countries of the former Soviet Union, though further work is required to determine the fitness effects of these SNPs and their prevalence in the region.”

Reviewer #2 (Remarks to the Author):

Please check for typo errors:

e.g. "These predictions aligned with phenotypic testing where available (Supplementary Table 1)." in the beginning of the Results section.

Thank you – we have removed typos.

Regarding the Results section, in the subsection named "Markers of clade expansion", would it be possible to add the values of LBIs when mentioning this index (e.g. adding the value of the mean LBI)?

Thank you for this suggestion. We have included the mean value of LBI for this tree:

“We found that the average LBI for a terminal node on the tree was 0.018 (range: 0.001, 0.037).”

REVIEWERS' COMMENTS

Reviewer #1 (Remarks to the Author):

The authors have responded to all the comments I raised.

I have no new comments